# Sex Differences in the Hypothalamic Oxytocin Pathway to Locus Coeruleus and Augmented Attention with Chemogenetic Activation of Hypothalamic Oxytocin Neurons

**DOI:** 10.3390/ijms22168510

**Published:** 2021-08-07

**Authors:** Xin Wang, Joan B. Escobar, David Mendelowitz

**Affiliations:** Department of Pharmacology and Physiology, School of Medicine and Health Sciences, The George University Washington, Washington, DC 20037, USA; xinwang@gwu.edu (X.W.); joan_escobar@gwu.edu (J.B.E.)

**Keywords:** hypothalamus, oxytocin, locus coeruleus, paraventricular nucleus, attention, sex differences, novel object

## Abstract

The tightly localized noradrenergic neurons (NA) in the locus coeruleus (LC) are well recognized as essential for focused arousal and novelty-oriented responses, while many children with autism spectrum disorder (ASD) exhibit diminished attention, engagement and orienting to exogenous stimuli. This has led to the hypothesis that atypical LC activity may be involved in ASD. Oxytocin (OXT) neurons and receptors are known to play an important role in social behavior, pair bonding and cognitive processes and are under investigation as a potential treatment for ASD. However, little is known about the neurotransmission from hypothalamic paraventricular (PVN) OXT neurons to LC NA neurons. In this study, we test, in male and female rats, whether PVN OXT neurons excite LC neurons, whether oxytocin is released and involved in this neurotransmission, and whether activation of PVN OXT neurons alters novel object recognition. Using “oxytocin sniffer cells” (CHO cells that express the human oxytocin receptor and a Ca indicator) we show that there is release of OXT from hypothalamic PVN OXT fibers in the LC. Optogenetic excitation of PVN OXT fibers excites LC NA neurons by co-release of OXT and glutamate, and this neurotransmission is greater in males than females. In male, but not in female animals, chemogenetic activation of PVN OXT neurons increases attention to novel objects.

## 1. Introduction

The tightly localized noradrenergic neurons (NA) in the locus coeruleus (LC) are well recognized as essential for focused attention, arousal, novelty-oriented responses and promoting wakefulness [1]. LC neurons possess slow irregular firing during quiet wakefulness which increases to sustained higher frequencies of firing during heightened vigilance and/or exposure to novel stimuli [2]. Children with autism spectrum disorder (ASD) exhibit diminished attention, engagement and orienting to exogenous stimuli, and it has been suggested that atypical LC-NA activity is involved in aberrant mental focus and responsiveness in children with ASD [3].

The oxytocin network within the CNS is known to play an important role in social behavior, pair bonding and cognitive processes in both sexes and is under investigation as a potential treatment for ASD [4]. In human subjects, oxytocin (OXT) administration enhances stimulus-induced pupil dilation, consistent with oxytocin augmenting attention towards socially relevant stimuli [5]. OXT administration in ASD children increases activity in brain regions important for perceiving social-emotional information [6]. Pharmacological and genetic (Oxtr-/-) inactivation of oxytocin receptor-mediated network function in mice diminishes the typical preference for novelty, not due to impairment in learning, but likely via decreased processing of novel contextual information [7]. Exposure to a social stimulus activates paraventricular nucleus of the hypothalamus (PVN) OXT neurons, and chemogenetic activation of PVN OXT of male mice enhances social investigation during a social choice test, while chemogenetic inhibition diminishes social preferences [8].

OXT is only synthesized in a limited number of discrete brain regions: the PVN, supraoptic and accessory nuclei of the hypothalamus [9]. While it is well known that PVN OXT neurons project to the LC in rats [10], and OXT fibers are present in the LC over its entire rostrocaudal extension in the human brain [11], little is known about the neurotransmission from PVN OXT neurons to LC NA neurons. In this study, we test, in male and female animals, whether PVN OXT neurons monosynaptically excite LC neurons, whether oxytocin is released and involved in this neurotransmission, and whether activation of PVN OXT neurons alters novel object recognition. Here, using oxytocin-sensitive sniffer cells, we show that there is release of OXT from hypothalamic PVN OXT fibers in the LC, optogenetic excitation of PVN OXT fibers excites LC NA neurons by co-release of OXT and glutamate, and finally chemogenetic activation of PVN OXT neurons increases attention assessed by novel object recognition tests. Surprisingly, there is significant sex dependent modulation in these responses.

## 2. Results

To test whether activation of PVN OXT neurons alters the response to a novel object, we selectively activated PVN OXT neurons using DREADDs in 10 male and six female animals. Chemogenetic activation of PVN OXT neurons in females did not change the proportion of time spent with a new object (see Figure 1, left), suggesting that PVN OXT neurons do not play an important role in this behavior in females. In contrast, however, chemogenetic activation of PVN OXT neurons in males significantly increased the time spent with a new, rather than familiar object (see Figure 1, right), indicating that PVN OXT neuron activation can facilitate attention to novel objects in male, but not female, rats.

To examine the neurotransmission from PVN OXT neurons to LC NA neurons, we selectively expressed ChR2 in PVN OXT neurons and their downstream fibers. Expression of ChR2 (green) in OXT neurons in the PVN was robust, as shown in Figure 2A, consistent with prior work [12]. Dense PVN OXT ChR2 fibers (green) were co-localized and often surrounded NA neurons shown using immunohistochemistry for dopamine β hydroxylase (blue) in the LC, Figure 2B. LC neurons that were recorded using patch clamp techniques were filled with biocytin (red), Figure 2C, and were subsequently confirmed to be LC NA neurons (merged image, Figure 2D).

Photoexcitation of ChR2 PVN OXT fibers in the LC evoked excitatory postsynaptic currents (EPSCs) in LC NA neurons. As shown in Figure 3A, each of the five consecutive photo-stimulations evoked a synaptic response in LC NA neurons. These responses were not altered by application of the GABA(A) receptor antagonist gabazine (*n* = 4, data not shown). The EPSCs were, however, significantly inhibited by application of the OXT receptor antagonist OTA, and were blocked by application of the glutamate receptor antagonists APV and CNQX (Figure 3A). The inhibition of photoactivated EPSCs by OXT and glutamate receptor antagonists was reversible (Figure 3A). We next tested whether there was any sex-dependent differences in the photoactivated EPSCs in LC NA neurons. Surprisingly, evoked EPSCs in males (15 cells from 15 slices) were significantly larger and nearly twice the amplitude of EPSCs elicited in females (11 neurons from 11 slices) (Figure 3B, first pulse evoked EPSC 44.4 ± 6 pA (F1) vs. 79 ± 12 pA (M1), *p* = 0.012; second pulse EPSC 31 ± 4 pA (F2) vs. 66 ± 9 pA (M2), *p* = 0.004; third pulse EPSC 29.4 ± 2 pA (F3) vs. 52.3 ± 7 pA (M3), *p* = 0.004; fourth pulse 26.1 ± 3 pA (F4) vs. 50 ± 9 pA (M4), *p* = 0.017; fifth pulse EPSC 23 ± 4 pA (F5) vs. 45 ± 6 pA (M5), *p* = 0.02. * *p* < 0.05, ** *p* < 0.01). To assess whether the larger EPSC amplitudes in males compared to females was accompanied by differences in OXT receptor activation, we examined the reduction in EPSC amplitude by the OXT receptor antagonist OTA. The reduction in EPSC amplitude with OTA was not different in males (10 neurons) compared to females (seven neurons, Figure 3B, right, percentage inhibition by OTA was −41.4 ± 7% (F1) vs. −44 ± 11% (M1); −55 ± 16% (F2) vs. −50 ± 11% (M2); −84 ± 15% (F3) vs. −48 ± 13% (M3); −66 ± 16% (F4) vs. −51 ± 12% (M4); −73 ± 15% (F5) vs. −32 ± 10% (M5)).

In response to a single 5 s duration photoexcitation of a spontaneous blue light, the frequency and amplitude of EPSCs significantly increased in LC NA neurons, Figure 3C (from 1.6 ± 0.3 Hz to 5.5 ± 0.9 Hz (females, *n* = 7, *p* = 0.004, F = 7.438), and 3.4 ± 0.7 Hz to 12 ± 2 Hz (males, *n* = 12, *p* = 0.0037, F = 6.923). The EPSC amplitude (D3) was enhanced from pre-stimulation of 17 ± 1.4 pA to 33 ± 5 pA (females, *n* = 7, *p* = 0.0027, F = 10.07), and 19 ± 3 pA to 32 ± 5 pA (males, *n* = 12, *p* = 0.0001, F = 16.45)). Furthermore, the increase in EPSC frequency was significantly greater in males than females (Figure 3C). However, OTA inhibited EPSC frequency similarly in males and females, and the increase in EPSC amplitude was similar in males and females (Figure 3C).

To assess whether either (1) the larger photo-evoked synaptic responses elicited by single pulses, or (2) the larger increase in EPSC frequency with a 5 s continuous pulse, which occurred in males compared to females, could be due to differences in OXT release, we utilized sniffer CHO cells to quantify OXT photo-released from ChR2-expressing PVN fibers in the LC in both five males and seven females. Optogenetic stimulation of PVN OXT fibers evoked large transient increases in Ca2+ within the sniffer CHO cells placed in close proximity to PVN OXT fibers in the LC, Figure 4. Surprisingly, the photostimulation-elicited increase in Ca2+ in the sniffer CHO cells upon PVN fiber activation was significantly less in males compared to females (23.6 ± 2.6%, *n* = 31 from 15 brain slices in seven females, 11.8 ± 2.3%, *n* = 21 from 12 slices in five males, *p* = 0.0028, unpaired *t*-test).

## 3. Discussion

The results in this study show that PVN OXT neuron activation increases attention to a novel object in male, but not in female, animals. In both male and female animals, PVN OXT fibers excite LC NA neurons. However, the evoked response from a single photoexcitation of PVN OXT fibers in LC NA neurons is greater in males than females, and the increase in EPSC frequency during a continuous 5 s stimulation is greater in males than females. The larger synaptic responses in males are unlikely to be due to a larger co-release of OXT from PVN OXT synaptic terminals, as sniffer cells for OXT release showed the opposite, with a greater levels of release OXT in females. In addition, the OXT receptor antagonist inhibited both synaptic responses to a single photoexcitation, and the increased frequency of EPSCs during a 5 s continuous photoactivation, similarly in males and females. As both control- and PVN OXT-elicited increases in EPSC amplitude were not different in males and females, it seems unlikely that there are any M/F differences in postsynaptic density of glutamate receptors in LC NA neurons. The increased evoked responses in males is likely due to greater facilitation of glutamate release onto LC NA neurons by co-released OXT. This is similar to the results in prior work that examined the effect of OXT on neurons in the nucleus tractus solitarius that receive vagal sensory input. OXT increased the amplitude of single evoked responses upon activation of vagal sensory fibers as well as the frequency of spontaneous EPSCs, but did not alter event kinetics or amplitudes of spontaneous EPSCs [13]. This suggests oxytocin release enhances glutamatergic release by as yet unknown mechanisms. One potential future avenue of investigation is the interaction between oxytocin and microglia, and, more specifically, whether oxytocin release alters the microglia-neuron microenvironment to facilitate excitation of downstream neuronal targets.

The sex-dependent results in rats in this study are supported by other studies that show male–female differences in OXT responses. In rats, intracerebroventricular (ICV) OXT administration enhanced social recognition and reversed social defeat-induced social avoidance in males [14] but not females [15,16]. Similarly, ICV OT facilitates social recognition and increases social approach (toward conspecifics that previously defeated them) in male but not female rats [14,15,16]. This work also provides a foundation for clinical studies that show sex differences in response to OXT. As examples, in a double-blinded crossover study intranasal (IN) OXT increased attention to feelings in young women and older men, but reduced it in older female participants [17]. In men, but not women, IN OXT increased emotional arousal and more positive behavior [18].

In conclusion, this work shows that PVN OXT neuron activation increases attention to a novel object in male, but not in female, animals. PVN OXT fibers excite LC NA neurons, and the magnitude of the excitatory neurotransmission is greater in males. The larger synaptic responses in males is unlikely due to differences in postsynaptic glutamate receptor density, or the amount of OXT released in this neurotransmission, but rather may be due to increased glutamate release in males compared to females.

## 4. Methods and Methods

### 4.1. Ethical Approval

All animal procedures were approved by the George Washington University Institutional Animal Care and Use Committee and in compliance with the panel of Euthanasia of the American Veterinary Medical Association and the National Institutes of Health (NIH) Guide for the Care and Use of Laboratory Animals. Experiments were conducted on 36 male and 34 female NIH SD rats from Hilltop.

### 4.2. Selective Expression and Activation of ChR2 and DREADDs in PVN OXT Neurons

Selective expression of either channelrhodopsin (ChR2) for in vitro optogenetic experiments, or excitatory Designer Receptors Exclusively Activated by Designer Drugs (DREADDs) for in vivo chemogenetic studies was accomplished using viral vectors in combination with the Cre-Lox system. As described previously, a rat minimal OXT promoter element from −530 bp to +33 bp relative to the origin of transcription of the OXT gene (UCSC genome browser on rat Nov. 2004 assembly; chr3:118,193,690 to 118,194,252) was synthesized de novo and flanked by multiple cloning sites (Genscript). The rAAV1-OXT-Cre was produced using the OXT promoter fragment pFB-AAV-OXT. The Cre promoter was created by cloning the OXT promoter into V032 by excising the OXT promoter/pUC57 with XbaI (5′) and AgeI (3′) and cloning it into V032 cut with SpeI (5′) and AgeI (3′). Then, Cre was added by cutting Cre out of pBS185 with XhoI (5′) and MluI-blunt (3′) and moving it into pFB-AAV-OXT cut with XhoI (5′) and Asp718-blunt (3′). To achieve robust and highly selective expression of DREADDs and ChR2 in PVN OXT neurons, the reporter viral vectors AAV2-hSyn-DIO-hM3D(Gq)-mCherry and excitatory ChR2 (AAV1-EF1a-DIO-hChR2) (UNC, Gene Therapy Center, Vector Core Services) were coinjected with AAV-OXT-Cre. Expression of these Cre-dependent vectors was only initiated in neurons selectively expressing Cre, as they contain silencing double-floxed inverse open reading frames. Previously published work showed, using immunohistochemical analysis, that this viral expression system elicited high (83.1 ± 2.1% and 93 ± 2.0%) selectivity for DREADDs and ChR2, respectively, in PVN OXT neurons [12,19].

Briefly, rat pups were anesthetized by hypothermia and placed on a stereotaxic frame with a neonatal adapter (Stoelting Co., Wood Dale, IL, USA). Stereotaxic coordinates for the PVN injection were calculated for the anterior-posterior plane by a correction factor (F = distance between bregma and lambda/3 = AP; ML ±0.3; DV-5.2). An amount of 30–50 nL of each virus combination was selectively microinjected over a 20 min period into the PVN as previously described [12]. Our previous work has demonstrated that injections of the DREADDs agonist, clozapine-N-oxide (CNO), increases the firing of PVN OXT neurons for at least one hour [12].

### 4.3. Optogenetic Experiments

Overall recordings were made from 42 LC neurons in 42 brainstem slices derived from 42 rats (21 rats for each sex) for optogenetic in vitro experiments. Rats at six to eight weeks old were anesthetized with isoflurane, sacrificed and transcardially perfused with ice-cold glycerol-based aCSF (in mM) (252 glycerol, 1.6 KCl, 1.2 NaH_2_PO_4_, 18 NaHCO_3_, 11 glucose, 1.2 MgCl_2_, 1.2 CaCl_2_, perfused with 95% O_2_ and 5% CO_2_, PH 7.4). The brain was carefully removed and brainstem slices (350 µm) were prepared by vibratome sectioning. The in vitro brainstem slices containing LC neurons were identified by their unique location where the upper roof of the 4th ventricle opens and the VII facial nerve fiber and superior medullary velum in the dorsal extent are present. Cut slices were then moved from the bath solution and incubated in the NMDG recovery solution (in mM) NMDG 93, HCl 93, KCl 2.5, NaH_2_PO_4_ 1.2, NaHCO_3_ 25, HEPES 20, D-Glucose 25, MgSO_4_ 10, CaCl_2_ 0.5 bubbled with 95% O_2_/5% CO_2_ at 34 °C in a water bath for 10 min. The brainstem slices were then moved to a recording chamber and perfused with standard aCSF solution contained (in mM) NaCl 125, KCl 3, NaHCO_3_ 25, HEPES 5, D-glucose 5, MgSO_4_ 1, CaCl_2_ 2 and continuously bubbled with 95% O_2_/5% CO_2_ to maintain pH at 7.4 at room temperature (22–24 °C). Whole cell patch clamp techniques were used to record photoactivated ChR2-evoked postsynaptic events in LC neurons. Biocytin (0.05%) was added in the patch solution to further identify the neurons in the LC using immunohistochemistry staining. The patch electrodes were filled with an intracellular recording solution at pH 7.3 containing (in mM): 135 cesium-methanesulfonate, 10 KCl, 10 HEPES, 1 MgCl_2_, 0.2 EGTA, 4 Mg-ATP, 0.3 GTP, 20 phosphocreatine. Lidocaine N-ethyl bromide (1 mg/mL) was included in the intracellular solution to block postsynaptic sodium currents. Photoactivated synaptic currents were elicited by photoactivation of ChR2 expressed in PVN OXT fibers with 5 pulses of blue light (at a frequency of 1 Hz, 1 msec duration, 3 mW optic power) or one continuous 5 s exposure (3 mW optic power) from a 473-nm laser (Crystal Laser) via a microscope objective. Synaptic events were detected using Clampfit 10.1 and Minianalysis (Synaptosoft, version 5.6.12).

Excitatory and inhibitory postsynaptic currents (EPSCs and IPSCs) were measured in voltage-clamp configuration. LC neurons were held at −55 mV to isolate glutamatergic synaptic transmission and record EPSCs, or +10 mV to isolate GABAergic synaptic transmission and record spontaneous IPSCs. Tetrodotoxin (TTX, 500 nM) and 4-Aminopyridine (4-AP, 100 µM) were included in the bath aCSF to isolate synaptic responses to monosynaptic neurotransmission. At the end of the electrophysiology experiments AMPA and NMDA glutamate receptors were inhibited by adding 6-cyano-7-nitroquinoxaline-2,3-dione (CNQX, 25 μM; Tocris) and D-2amino-5-phosphopentanoic acid (AP5; 50 µM, Tocris) into ACSF, and GABA(A) receptors were inhibited by gabazine (25 µM, Tocris). The specific OXT receptor antagonist, d(CH_2_)_5_^1^,Tyr(Me)^2^,Thr^4^,Orn^8^,des-Gly-NH_2_^9^)-Vasotocin trifluoroacetate salt (1 µM, Bachem H2908) was used to block OXT receptors. Focal drug application was performed using a PV830 Pneumatic PicoPump pressure delivery system (WPI, Sarasota, FL, USA) to apply drugs from a patch pipette positioned within 30 μm from the patched LC neuron.

The cells from which recordings were obtained were confirmed as LC NA neurons by immunohistochemistry. After the electrophysiological experiments, the tissue was fixed in 4% paraformaldehyde overnight, then mounted and cover slipped with Prolong anti-fade mounting medium (Invitrogen, Eugene, OR). The tissue was processed for dopamine β hydroxylase using the following primary antibodies (overnight incubation at 22–24 °C): mouse anti-Dopamine β Hydroxylase antibody (1:1000 dilution; MAB 308, Millipore, MA, USA), rabbit anti-GFP/EYFP (1:500 dilution; Abcam, Cambridge, MA, USA). Secondary antibodies were goat anti-mouse Alexa Fluor 405 and goat anti-rabbit Alexa Fluor 488 (all 1:200 dilution and 4 h incubation at 22–24 °C; Jackson ImmunoResearch, West Grove, PA, USA). Confocal stack images were collected with 10× and 63× objectives of a Zeiss 710 confocal system.

### 4.4. Sniffer CHO Cells for OXT

Sniffer CHO cells were used to examine release of OXT in the LC upon optogenetic activation of ChR2-expressing PVN OXT fibers. OXT receptors, as well as the red fluorescent calcium indicator, R-GECO1, were expressed in CHO cells as previously described [20]. In brief, CHO cells were transfected with pcDNA3.1+ containing human OXTr cloned in at EcoRI (5′) and XhoI (3′) (plasmid obtained from Missouri S&T cDNA Resource Center; www.cdna.org) using lipofectamine, and stable over-expression was achieved by genetcin (500 µg/mL) selection. OXTr-expressing CHO-cells were then plated and transiently transfected to also express the red fluorescent genetically encoded Ca2+ indicator (R-GECO; plasmid kindly donated by Robert Campbell, University of Alberta, Canada; Addgene plasmid 32444) with Fugene 6. OXT receptor-expressing sniffer CHO cells were pipetted onto slices that contained the LC. Imaging was performed on a confocal microscope system consisting of an upright Zeiss Axio Examiner Z1 microscope, with a W Plan Apocromat 20×/1.0 objective, equipped with Carl Zeiss 710 confocal hardware. Z-series spectral image sets were used to produce two channel image sets representing ChR2-EYFP fibers and sniffer cells, by applying off-line a linear spectral un-mixing protocol. For Ca2+ imaging upon photo-excitation of the ChR2 fibers, images measured 128 × 128 pixels taken at 2.3 zoom factor and bi-directional scanning. Thus, the pixel measured 1.44 µm, providing sufficient cellular and temporal resolution. Images were obtained every 76 msec.

### 4.5. In Vivo Novel Object Recognition

Attention in response to a novel stimulus was assessed using automated behavioral analysis systems (HomeCageScan 3.0 and Capture Star software, Version 1; CleverSys Inc., Reston, VA, USA). Animals were acclimated with an intraperitoneal injection (i.*p*.) with 250 microliters of normal saline solution and placement of the animal home cage in the observation system for 3 h each day for 3 days prior to the day of the first experiment. On the day of the first experiment, animals randomly received either an i.*p*. injection of saline, or CNO (1 mg/kg) to activate PVN OXT neurons. One hour post injection, two different objects were randomly selected from a pool of objects similar in size but different in shape and color, and were placed at each end of the cage. Both objects were removed after 15 min, and animals were left for another hour with no objects. One of the objects from the initial set and one randomly selected novel object was then placed in the cage for 15 min and the proportion of time the animal interacted with the novel object was quantified. Experiments were repeated the next day with different objects and with the alternate injection (CNO or saline).

### 4.6. Experimental Design and Statistical Analyses

All electrophysiological data were collected and digitized using Clampex (10.2, Molecular Devices, Sunnyvale, CA, USA) and analyzed with Clampfit (10.2, Axon Instrument, Molecular Devices, LLC., San Jose, CA, USA). Photoactivating ChR2 evoked postsynaptic events were detected using MiniAnalysis (version 6.0.7 Synaptosoft, Decatur, GA, USA) with the minimal acceptable amplitude at 10 pA. Synaptic currents containing one continuous 5 s photoactivation were grouped into 13 consecutive 1 s epochs. The first 3 s before photo stimulation were control (pre-stim), followed by five 1 s continuous epochs (stimulation) and a 5 s recovery (post-stim). All data were within a normal distribution with equal variances using frequency distribution and Bartlett’s corrected statistic test, respectively. Statistical analysis and generation of scatter plots and histograms were performed using Graphpad Prism 5.0 (Graphpad Software, San Diego, CA, USA), Microcal Origin 6.1 (OriginLabs, Northhampton, MA, USA), and Microsoft Excel (Microsoft, Redmond, WA, USA). Data are presented as mean ± SEM, and for female versus male comparisons we used unpaired Student’s *t*-tests. For within group comparisons prior to, during, and post photoexcitation, we used one-way ANOVA with repeated measures (RM) and Dunnett’s post-hoc analysis. Two-way RM ANOVA with Bonferroni post tests were applied for between female and male group comparisons. For all comparisons the significance level was set at *p* < 0.05. **** *p* < 0.0001; *** *p* < 0.001; ** *p* < 0.01; * *p* < 0.05.

## 5. Conclusions

This work shows that there are very significant sex differences in the increased attention to novel objects that occur with chemogenetic activation of hypothalamic oxytocin neurons. We also show sex differences in the photoactivated neurotransmission from oxytocin neurons to the locus coeruleus, a brain region essential for cognitive focus and attention.

## Figures and Tables

**Figure 1 ijms-22-08510-f001:**
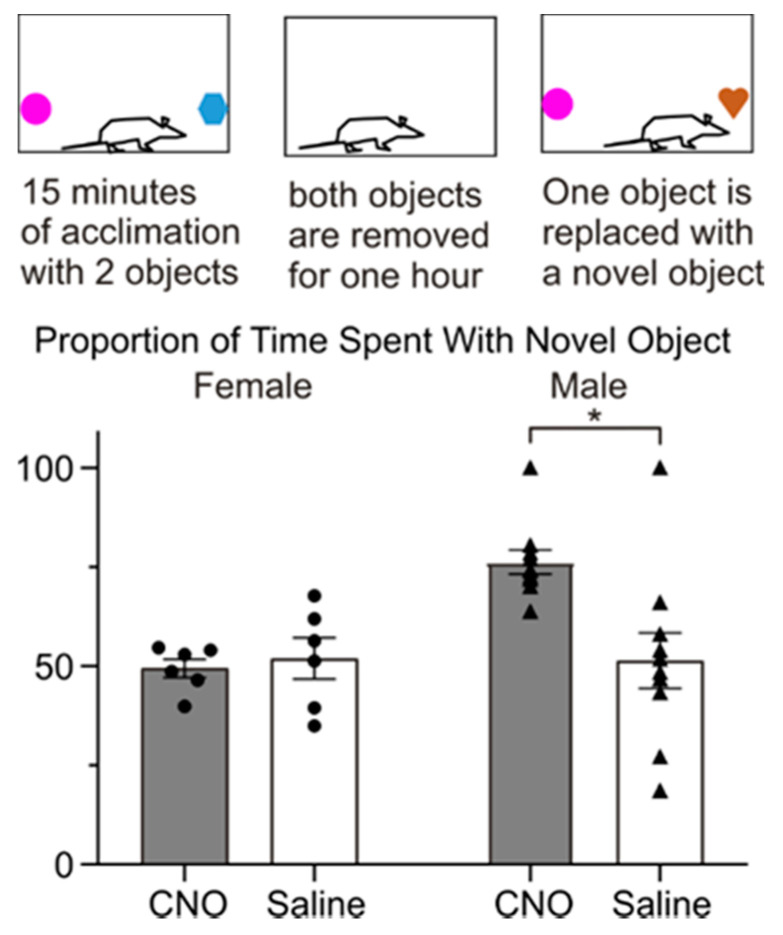
Male rats spent more time interacting with a novel subject than female rats. A graphic representation of the novel object paradigm is shown (**top**). Animals received an injection of Clozapine-N-Oxide (CNO) or saline 1 h prior to acclimation. Two randomly selected objects were placed within the cage and the animal was allowed to interact with both objects for 15 min. Both objects were removed for one hour, and then one of the former objects, along with one novel object was placed in the cage for 15 min. Histogram (**bottom**) shows the proportion of time interacting with a novel object (filled bars are with CNO treatment, open bars were with saline injection). While activation of DREADDs expressing PVN OXT neurons with CNO had no effect in females, activation of PVN OXT neurons significantly increased the proportion of time spent with the novel object in males. * *p* < 0.05.

**Figure 2 ijms-22-08510-f002:**
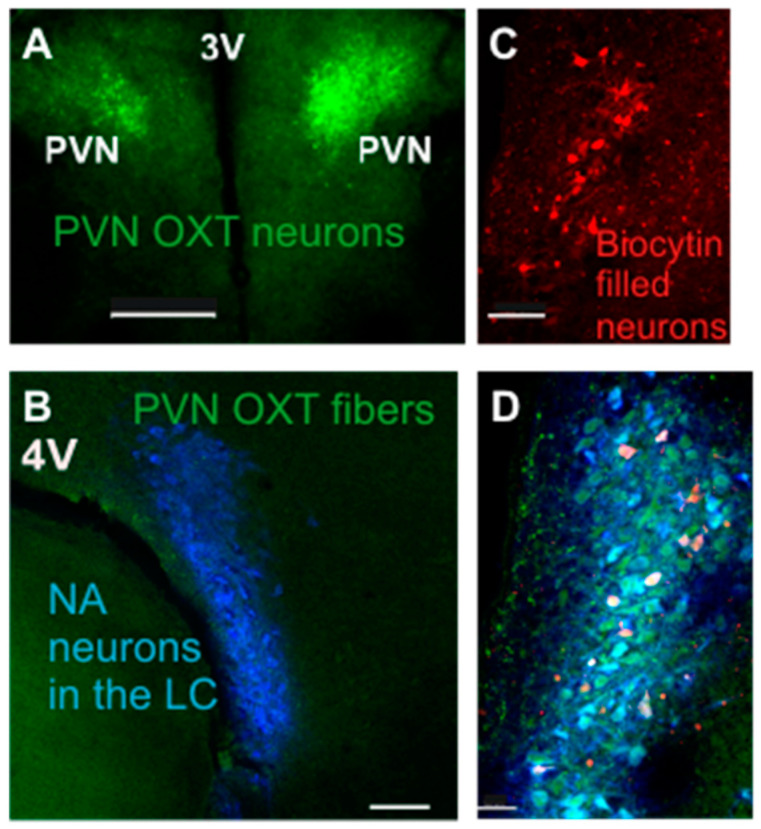
Selective expression of ChR2 in PVN OXT neurons and colocalization of ChR2 fibers surrounding DβH+ and biocytin filled NA neurons in the locus coeruleus. (**A**) Low power confocal image shows ChR2-EYFP (green) expression in the PVN. (**B**) PVN OXT ChR2 fibers (green) co-localize and surround LC noradrenergic neurons (NA) shown with dopamine β hydroxylase (DβH) immunohistochemistry (blue). (**C**) LC NA neurons recorded with patch clamp electrodes were filled with biocytin (red); (**D**) Merged image illustrates the co-localization of PVN OXT fibers (green) with DβH LC NA neurons (blue), and biocytin-filled patched LC NA neurons (red). Scale bar 400 µm (**A**), 200 µm (**B**), 100 µm (**C**), and 50 µm (**D**).

**Figure 3 ijms-22-08510-f003:**
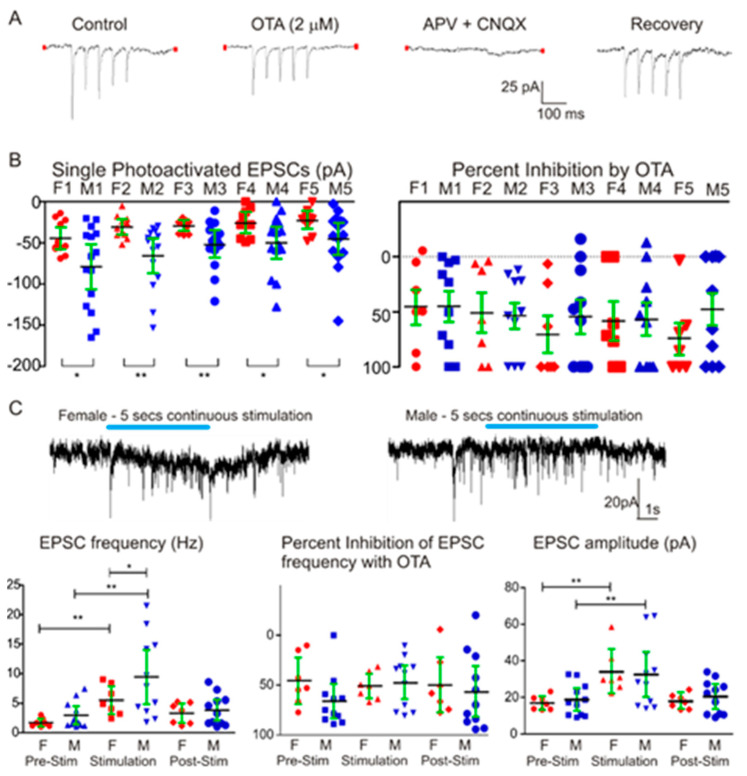
Photoactivation of PVN OXT ChR2 expressing fibers elicited postsynaptic excitatory currents in LC NA neurons that were greater in male compared to female animals. (**A**) Representative traces recorded in voltage clamp configuration from a LC NA neuron. The specific oxytocin receptor antagonist, OTA inhibited, while the AMPA and NMDA glutamate receptor antagonists—APV and CNQX, respectively—blocked photoactivated synaptic responses. This inhibition was reversible following washout of OTA, APV and CNQX. (**B**) The photostimulated EPSCs were significantly greater in males (*n* = 15) than females (*n* = 11) in all five consecutive photoactivations (five pulses, 1 msec/pulse, left). The percentage inhibition of evoked EPSCs by the oxytocin receptor antagonist, OTA, (**B**, right) were not different between males and females. (**C**) Representative synaptic events prior to, during, and following a five second continuous photoactivation of ChR2 fibers in female (left) and male (right) LC NA neurons. EPSC frequency increased significantly in both males and females during the 5 s photostimulation. The facilitation of EPSC frequency was significantly greater in male than in female animals. OTA significantly inhibited the increase in EPSC frequency during the photoactivation, and this was not different in males compared to females. EPSC amplitude also increased during the photoactivation, and this augmentation of EPSC amplitude was similar in males and females. ** *p* < 0.01; * *p* < 0.05.

**Figure 4 ijms-22-08510-f004:**
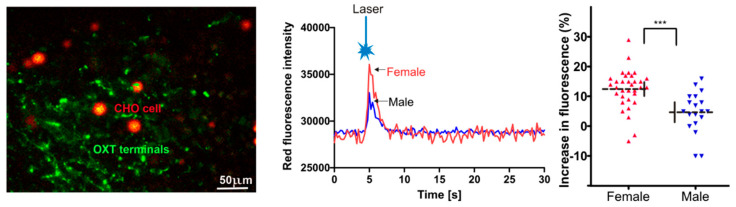
Sniffer CHO cells show that the release of OXT from PVN OXT fibers in the LC is greater in females than males. A representative confocal image showing sniffer CHO cells expressing OXT receptors and the Ca indicator R-GECO1 (red) in close apposition to PVN OXT ChR2-EYFP fibers (green). Scale bar 50 µm (**left**). A single laser pulse photo-activated ChR2 fibers and increased R-GECO1 fluorescence in neighboring sniffer CHO cells (**middle**). The responses were significantly greater in females than in males (**right**). *** *p* < 0.001.

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
