# Peer review of "Sex Differences in the Hypothalamic Oxytocin Pathway to Locus Coeruleus and Augmented Attention with Chemogenetic Activation of Hypothalamic Oxytocin Neurons"

_ijms, 2021, doi:10.3390/ijms22168510_

Round 1

Reviewer 1 Report

The authors showed sex differences in synaptic transmission from PVN oxytocin neurons to the noradrenergic neurons in the LC using AAV-pOT-Cre and AAV-DIO-ChR2 in rats. They also used oxytocin sensitive sniffer cells and found that the responses were significantly greater in female rats than male rats. In addition, they showed that chemogenetic activation of PVN oxytocin neurons increases time spent with the novel object only in male rats. This study provides novel and interesting findings on neural transmission, but I have several concerns about some parts of the experimental design. The authors should describe the methods in detail. I make some comments below to improve the manuscript.

Major comments:

  1. The specificity of the expression induced by AAV vectors

The authors used AAV vectors to induce specific expression of ChR2 or hM3Dq in PVN oxytocin neurons. Almost all the data in the manuscript rely on the specificity of oxytocin promoter used in AAV vectors. Generally, it is difficult to induce specific expression by using a short promoter that can be contained within AAV vectors. In addition, even small leak expression of Cre recombinase can induce recombination and results in plenty amount of expression of DIO-ChR2 or DIO-hM3Dq. The authors should show the data of colocalization of oxytocin and ChR2 in the AAV injected rats. They cite their previous papers using similar methodology, but they are not enough.

  1. Novel object recognition tests

In Figure 1 the authors showed the results on novel object recognition tests. Figure 1 shows that both male and female rats have no preference for novel objects in saline-injected condition, is it true? Normally, in novel object recognition tests rats show preference for novel objects. Therefore, I am not confident that these experimental procedures worked well.

  1. Axonal terminal and neuropeptide secretion

The authors investigate the synaptic transmission from PVN oxytocin neurons to the NA neurons in the LC. They showed in Figure 2 that dense fibers from PVN oxytocin neurons in the LC. However, the LC is not recognized as major projection sites of PVN oxytocin neurons. Even the paper the author cited (Sofroniew, 1980) showed that oxtyocin fibers from the PVN in the LC is indicated as (+), while that in the NTS is indicated as (++++). The authors should show immunostaining of oxytocin in the LC region. It is common that some of axonal projections do not include neuropeptides. Therefore, comparing the localization of axonal fibers and oxytocin might be important to think about the synaptic transmission in the LC.

  1. Methodology in detail

Methods are insufficiently described throughout the manuscript. Please refer to “Minor comments” below.

Minor comments:

p2, line 55-56

In this study we test, in male and female animals, if PVN OXT neurons monosynaptically excite LC neurons

If oxytocin mediates neuronal transmission between PVN oxytocin neurons and NA neurons in the LC, volume transmission can be involved. Therefore, the word “monosynaptic” is not adequate.

p2, line 58

Here, using oxytocin sensitive sniffer cells, we show that there is endogenous release of OXT from hypothalamic PVN OXT fibers,

Please explain sniffer cells briefly here.

p3, line 84-85

To examine the neurotransmission from PVN OXT neurons to LC NA neurons we selectively expressed ChR2 in PVN OXT neurons and their downstream fibers.

Show quantitative data to show that ChR2 are selectively expressed in PVN oxytocin neurons. Show immunostaining data to confirm colocalization of oxytocin and ChR2 in the PVN and in the LC. Describe the percentage of colocalization in the PVN.

p3, line 92

The legends of the scale bars in Figure 2 are not unreadable. They should be removed or modified.

p5, line 161

Sniffer CHO cells show endogenous release of OXT from PVN OXT synaptic endings in the LC is greater in females than males.

The word “endogenous” is not adequate because it is “evoked” by optogenetic stimulation. Another concern is the source of oxytocin. What kind of data confirmed that oxytocin is released only from “synaptic endings”?

p6, line 215-216

Cre recombinase was exclusively driven by the OXT promoter (rAAV1-OXT-Cre)

As I pointed out above, the specificity of the OXT promoter is critically important in this study. Generally, it is difficult to induce specific expression by using a short promoter that can be contained within AAV vectors. Please add references for the oxytocin promoter used in this study. I presume that this OXT promoter is that used in (Knobloch, 2012-Neuron). Is it correct?

p6, line 218

(AAV1-EF1a-DIO-hChR2) or floxed excitatory DREADDs (AAV2-hSyn-DIO-hM3D(Gq).

Please describe the virus vectors in detail. What kind of ChR2 is used? Not fused with fluorescent proteins? What is the titer of virus vectors? How did the authors get the virus vectors? Important information should be described in the manuscript even when the related papers are cited.

p7, line 251-254

Photoactivated synaptic currents were elicited by photoactivation of ChR2 expressed in PVN OXT fibers with 5 pulses of blue light (at a frequency of 1Hz, 1 msec duration, 3mW optic power) or one continuous 5 sec exposure 253 (3mW optic power) from a 473-nm laser (Crystal Laser) via a microscope objective.

Describe the approximate size of the blue light illumination area. Considering volume transmission of oxytocin, this information can be important.

p7, line 283

The authors should describe the methods in detail. For readers who are not familiar with sniffer cells some review papers should be cited.

Author Response

We would like to thank both reviewers for their very thorough and constructive review.

R#1

“This study provides novel and interesting findings on neural transmission, but I have several concerns about some parts of the experimental design. The authors should describe the methods in detail”.

We would like to thank the reviewer for recognizing the interesting and novel findings in this work. As requested by the reviewer, we have now revised the manuscript to include new and more detailed methods. 

  1. The specificity of the expression induced by AAV vectors

We now state in the methods “As described previously, a rat minimal OXT promoter element from −530 bp to +33 bp relative to the origin of transcription of the OXT gene (UCSC genome browser on rat Nov. 2004 assembly; chr3:118,193,690 to 118,194,252) was synthesized de novo and flanked by multiple cloning sites (Genscript). The rAAV1-OXT-Cre was produced using the OXT promoter fragment pFB-AAV-OXT. The Cre promoter was created by cloning the OXT promoter into V032 by excising the OXT promoter/pUC57 with XbaI (5′) and AgeI (3′) and cloning it into V032 cut with SpeI (5′) and AgeI (3′). Then Cre was added by cutting Cre out of pBS185 with XhoI (5′) and MluI-blunt (3′) and moving it into pFB-AAV-OXT cut with XhoI (5′) and Asp718-blunt (3′). To achieve robust and highly selective expression of DREADDs and ChR2 in PVN OXT neurons, the reporter viral vectors AAV2-hSyn-DIO-hM3D(Gq)-mCherry and excitatory ChR2  (AAV1-EF1a-DIO-hChR2)  (UNC, Gene Therapy Center, Vector Core Services) were coinjected with AAV-OXT-Cre. Expression of these Cre-dependent vectors was only initiated in neurons selectively expressing Cre, as they contain silencing double-floxed inverse open reading frames. Previously published work showed, using immunohistochemical analysis, this viral expression system elicited high (83.1 ± 2.1% and 93 ± 2.0 %) selectivity for DREADDs and ChR2, respectively, in PVN OXT neurons (Heather Jameson et al., 2016; Ramón A Piñol et al., 2012).”

It is also worth noting that this manuscript was submitted for review and potential publication in a Special Issue entitled "Therapeutic Potential of Targeting the Oxytocinergic System", to be published in the International Journal of Molecular Sciences. There is not sufficient time for additional experiments given the deadline of this Special Issue.

  1. Novel object recognition tests

We agree with the reviewer that a 50-55% increase in time spent with the novel object in females, and males given saline, is lower than expected.  This might be due our experimental paradigm that included an injection of saline or CNO prior to the test.  There may have been larger increases in time spent with the novel object if this test was performed without an injection.  However this injection was unavoidable and necessary to test our hypothesis that activation of DREADDs in PVN OXT neurons alters time spent with a novel object, and since this protocol, including an injection, was used in all 4 groups it does not interfere with our comparison between groups, and interpretation of the data that activation of PVN OXT neurons in males, but not females, increases time spent with a novel object.

  1. Axonal terminal and neuropeptide secretion

We agree with the reviewer that other work has shown the pathway from PVN OXT neurons to the LC is not as strong as other pathways (such as the PVN OXT pathway to the NTS). However, we respectively note that the density of visualized oxytocin fibers using immunohistochemistry may not correlate with the strength of  neurotransmission in any given pathway in-vivo or in-vitro – especially since in this pathway, and many others from PVN OXT neurons, PVN OXT fibers co-release glutamate, and this glutamatergic release is a major determinant of the strength of this neurotransmission.

Minor Comments

If oxytocin mediates neuronal transmission between PVN oxytocin neurons and NA neurons in the LC, volume transmission can be involved. Therefore, the word “monosynaptic” is not adequate.

We agree with the reviewer and delete the word “monosynaptic”

Please explain sniffer cells briefly here.

We now include the following description “Using “oxytocin sniffer cells” (CHO cells that express the human oxytocin receptor and a Ca indicator)”

Show quantitative data to show that ChR2 are selectively expressed in PVN oxytocin neurons. Show immunostaining data to confirm colocalization of oxytocin and ChR2 in the PVN and in the LC. Describe the percentage of colocalization in the PVN.

We now more fully describe in the methods, using identical viruses, animal strains and ages, “using immunohistochemical analysis, this viral expression system elicited high (83.1 ± 2.1% and 93 ± 2.0 %) selectivity for DREADDs and ChR2, respectively, in PVN OXT neurons (Heather Jameson et al., 2016a; Ramón A Piñol et al., 2012).”

The legends of the scale bars in Figure 2 are not unreadable. They should be removed or modified.

As suggested by the reviewer, they have been removed.

The word “endogenous” is not adequate because it is “evoked” by optogenetic stimulation. Another concern is the source of oxytocin. What kind of data confirmed that oxytocin is released only from “synaptic endings”?

As suggested by the reviewer, the words “endogenous” and “synaptic endings” have been removed.

Describe the approximate size of the blue light illumination area.

We are highly reluctant to approximate the size of the illumination area since whatever we approximate could be inaccurate.  In future experiments, we will strive to quantify the area of ChR2 excitation, but of course, this area depends not only on the intensity of the blue light but also the expression levels of ChR2.

The authors should describe the methods in detail. For readers who are not familiar with sniffer cells some review papers should be cited.

More methods concerning sniffer cells are now included.  We now state ”In brief, CHO cells were transfected with pcDNA3.1+ containing human OXTr cloned in at EcoRI (5′) and XhoI (3′) (plasmid obtained from Missouri S&T cDNA Resource Center; www.cdna.org) using lipofectamine and stable over-expression was achieved by genetcin (500 µg/ml) selection. OXTr-expressing CHO-cells were then plated and transiently transfected to also express the red fluorescent genetically encoded Ca2+ indicator (R-GECO; plasmid kindly donated by Robert Campbell, University of Alberta, Canada; Addgene plasmid 32444) with Fugene 6. OXT receptor-expressing sniffer CHO cells were pipetted onto slices that contained the LC.  Imaging was performed on a confocal microscope system consisting of an upright Zeiss Axio Examiner Z1 microscope, with a W Plan Apocromat 20x/1.0 objective, equipped with Carl Zeiss 710 confocal hardware. Z-series spectral image sets were used to produce two channel image sets representing ChR2-EYFP fibers and sniffer cells, by applying off-line a linear spectral un-mixing protocol. For Ca2+ imaging upon photo-excitation of the ChR2 fibers, images measured 128×128 pixels taken at 2.3 zoom factor and bi-directional scanning. Thus the pixel measured 1.44 µm, providing sufficient cellular and temporal resolution. Images were obtained every 76 msec.”

Reviewer 2 Report

This article aims to discuss the sex differences in the neurotransmission of oxytocin in the hypothalamic pathway and the divergent effects in attention to novel objects by chemogenetic activation of paraventricular oxytocin neurons in rats.

The manuscript is written comprehensively enough but still a bit difficult to be understandable for the general audience.

Sex differences in the oxytocinergic system have been discussed since the 1980s. However, what is new evidence that can be learned from this study and the further implications for future research? It would be great if the authors could discuss more regarding this issue.
1. The hypothalamic periventricular nucleus is a place wherein many neurotransmitters with sex-specific roles are synthesized. Is that the reason for the sex differences in neurotransmission and chemogenetic activation?
2. More oxytocin neurons are found in the hypothalamic paraventricular nucleus in male rats than the female rats. Is that the possible reason for the different levels of chemogenetic activation?

The authors stated that "Children with autism spectrum disorder exhibit diminished attention, engagement and orienting to exogenous stimuli" and suggested a correlation with "the attention to novel objects" in this study. Further, the authors wrote that in the conclusion "This work ties together three important areas of research; the role and modulation of the LC in attention and wakefulness, the mechanisms by which oxytocin changes behavior, mental focus and can be a potential treatment for ASD". The use of sweeping statements (e.g. wakefulness, attention to ASD) is another concern.

Please check the typos in the figure legend of Figure 1 and correct the indentation of the figure legend of Figure 2. 

Author Response

Reviewer #2

We would like to thank reviewer #2 for his/her thorough and constructive review.

“What is new evidence that can be learned from this study and the further implications for future research? It would be great if the authors could discuss more regarding this issue.”

As suggested by the reviewer we now have expanded our discussion, and now propose: “This suggests oxytocin release enhances glutamatergic release by yet unknown mechanisms.  One potential future avenue of investigation is the role of oxytocin and microglia, and, more specifically, if oxytocin release alters the microglia-neuron microenvironment to facilitate excitation of downstream neuronal targets.”

  1. The hypothalamic periventricular nucleus is a place wherein many neurotransmitters with sex-specific roles are synthesized. Is that the reason for the sex differences in neurotransmission and chemogenetic activation?

While further discussions of other neurotransmitters in the PVN, and their sex-differences, are beyond the scope of this study, we agree with the reviewer that neurons in the paraventricular nucleus of the hypothalamus likely play a major role in both well- known and yet to be elucidated sex-dependent functions and activities.

  1. More oxytocin neurons are found in the hypothalamic paraventricular nucleus in male rats than the female rats. Is that the possible reason for the different levels of chemogenetic activation?

While we did not explore this possibility directly, our results do not suggest this is a likely explanation for the results in this study as sniffer cells for OXT release showed greater levels of OXT release in females compared to males.

The authors stated that "Children with autism spectrum disorder exhibit diminished attention, engagement and orienting to exogenous stimuli" and suggested a correlation with "the attention to novel objects" in this study. Further, the authors wrote that in the conclusion "This work ties together three important areas of research; the role and modulation of the LC in attention and wakefulness, the mechanisms by which oxytocin changes behavior, mental focus and can be a potential treatment for ASD". The use of sweeping statements (e.g. wakefulness, attention to ASD) is another concern.

We have edited these statements to be less sweeping.

Please check the typos in the figure legend of Figure 1 and correct the indentation of the figure legend of Figure 2.

Typos have been corrected.

Round 2

Reviewer 1 Report

The authors’ responses to my comments were sincere and generally convincing. The manuscript has been effectively improved.

Author Response

Thank you.

Reviewer 2 Report

In Figure 1, please correct the "Male rates" to "Male rats". The description below the figures and the Figures Legend at the end of the manuscript are mismatched, please check for consistency.

In the revised manuscript, the authors have addressed my concerns. I feel that the paper is a valuable addition to the literature. I have no further additional comments.

Author Response

These issues have been corrected. Thank You.